# Development of an expected possession value model to analyse team attacking performances in rugby league

Thomas Sawczuk[1,2,3]*, Anna Palczewska[1], Ben Jones[2,3,4,5,6]

**1** School of Built Environment, Engineering and Computing, Leeds Beckett University, Leeds, United Kingdom, **2** Carnegie Applied Rugby Research (CARR) Centre, Carnegie School of Sport, Leeds Beckett University, Leeds, United Kingdom, **3** England Performance Unit, The Rugby Football League, Red Hall, Leeds, United Kingdom, **4** Leeds Rhinos Rugby Club, Headingley Carnegie Stadium, Leeds, United Kingdom, **5** Division of Exercise Science and Sports Medicine, Department of Human Biology, Faculty of Health Sciences, The University of Cape Town and the Sports Science Institute of South Africa, Cape Town, South Africa, **6** School of Science and Technology, University of New England, Armidale, New South Wales, Australia

\* t.sawczuk@leedsbeckett.ac.uk

**Data Availability Statement:** The data that was used for this study was acquired from a third-party, formerly Opta Sports, now Stats Perform. It is available from www.optaprorugby.com. The data was provided under a license agreement with Opta

## Abstract

This study aimed to evaluate team attacking performances in rugby league via expected possession value (EPV) models. Location data from 59,233 plays in 180 Super League matches across the 2019 Super League season were used. Six EPV models were generated using arbitrary zone sizes (EPV-308 and EPV-77) or aggregated according to the total zone value generated during a match (EPV-37, EPV-19, EPV-13 and EPV-9). Attacking sets were considered as Markov Chains, allowing the value of each zone visited to be estimated based on the outcome of the possession. The Kullback-Leibler Divergence was used to evaluate the reproducibility of the value generated from each zone (the reward distribution) by teams between matches. Decreasing the number of zones improved the reproducibility of reward distributions between matches but reduced the variation in zone values. After six previous matches, the subsequent match's zones had been visited on 95% or more occasions for EPV-19 (95±4%), EPV-13 (100±0%) and EPV-9 (100±0%). The KL Divergence values were infinity (EPV-308), 0.52±0.05 (EPV-77), 0.37±0.03 (EPV-37), 0.20±0.02 (EPV-19), 0.13±0.02 (EPV-13) and 0.10±0.02 (EPV-9). This study supports the use of EPV-19 and EPV-13, but not EPV-9 (too little variation in zone values), to evaluate team attacking performance in rugby league.

## Introduction

In recent years, the growing availability of event level data in rugby league has led to an increase in research surrounding the characteristics of match winning performances [1–4]. These studies can broadly be split into two categories based on the inclusion of spatial (i.e. event location) data within their analyses [3], or lack thereof [1, 2, 4]. Although those studies not including spatial data provide valuable insights into potential match winning actions [2] or

Sports/Stats Perform, and the data is subject to an approved research ethics application from our University. The terms of our license agreement prevent us from sharing the raw data we used for this analysis. Our ethical approval also prevents us from sharing any data in any way that could be re-identified. The metadata and (fixture/location/action) data itself would allow someone else to re-identify fixtures, teams and/or players, breaching the ethics approval given. However, it should be possible to obtain access to the data by contacting Stats Perform (www.statsperform.com/contact/). The authors had no special access privileges and all data is taken from the 2019 season of the Super League. The corresponding author is happy to liaise with any researchers who have queries about how to obtain the data.

**Funding:** The author(s) received no specific funding for this work.

**Competing interests:** The authors have declared that no competing interests exist.

the classification of player positions [4], they do not account for some of the most valuable contextual information surrounding the location of the events analysed. Incorporating this spatial context into the analysis of a team's attacking performances could have a significant impact on tactical preparations for future matches and thus provides a valuable avenue for future research in rugby league.

Spatial data has been included in analyses of team and player performances via expected possession value (EPV) models across several sports [3, 5, 6]. EPV models assign a value to every pitch location (or location-action tuple) visited during a match based on the probability of scoring a goal, basket or try from that action/location within a given amount of time. These values can be summed at a match level to quantify attacking performances. EPV models have been developed using either probabilistic [5, 7] or stochastic [8, 9] methods. Fernandez et al. [5, 7] used probabilistic deep learning methods to establish the EPV in soccer, whereas Routley and Schulte [9] and Liu and Schulte [8] both used stochastic models based upon Markovian principles to evaluate ice hockey player performances via Q-values. The episodic nature of rugby league, whereby teams have a finite six tackle period of possession, barring fouls/errors, ensures it is perfectly suited to stochastic analyses. Despite this, limited research has been conducted in rugby league using stochastic analyses [3].

When considering spatial data within EPV models, it is common to split the pitch into different zones, which pool data together. Discretising the pitch into these zones is computationally efficient and allows for improved generalisability to other samples. Nevertheless, selecting the correct zone system is fraught with difficulty. If the zones are too large, valuable data could be lost, but if they are too small, the results of the analysis will not be generalisable [10]. Two key methods have been used to determine zone sizes within the EPV literature. These are fixed zone sizes, arbitrarily chosen by the authors [3], and the selection of zones based on likely shooting locations [6, 8]. Given a try can be scored across the width of the pitch, it isn't possible to identify specific shooting (or try scoring) locations within rugby league. However, it may be possible to produce a set of zones which are more rugby league specific by aggregating different zones based on their point scoring potential. This aggregated set of zones may prove more adept at evaluating attacking performances within rugby league than the fixed size zones previously considered [3], but to date no study has compared these methods.

Within a model evaluating rugby league attacking performances, one of the most important elements is the reproducibility of performances between fixtures [11]. To understand the reproducibility of attacking performances from the perspective of an EPV model, it is necessary to evaluate the similarity of the total EPV generated during a match by a zone (i.e. the zone's match EPV) between fixtures. Completing such an analysis with EPV models using different zone sizes (e.g. the previously published fixed zone size [3], a smaller fixed zone size and aggregated zone sizes) would help to identify the most suitable set of zones for rugby league.

The aims of this study were to i) produce six EPV models (two with fixed zone sizes of ~5m x 5m and ~10m x 10m [3], and four with aggregated zones based on differences in the zones' match EPV of 0.5, 1.0, 1.5 and 2.0 points per match) to quantify expected points scored during rugby league matches from specific locations on the pitch (i.e. attacking performances), ii) compare the reproducibility of match EPV between fixtures for the EPV models, and iii) quantify individual teams' attacking performances across a season using an EPV model. Six EPV models were chosen to provide a range of zone sizes from small to large, which could be compared with regards to their reproducibility between matches and usefulness in practice.

## Methods

### Sample

Event level match-play data were obtained from Opta (Stats Perform, London, UK) for all 180 matches of the 2019 Super League season. In total, 59,233 plays were analysed. Within this sample, 1,369 tries were scored (1,013 successful conversions, 356 unsuccessful conversions), 271 penalty goals were attempted (239 successful, 32 unsuccessful) and 89 drop goals were attempted (42 successful, 47 unsuccessful). Prior to analysis, informed consent was obtained and ethics approval was provided by a University sub-ethics committee.

### Data pre-processing

Although event level data can include various pieces of information regarding the actions completed and players involved, Opta only includes location data (x and y co-ordinates) for the first action of each play. Consequently, only the location of the first action of each play was used when developing the EPV models. Despite some variation being present in pitch sizes across the Super League, Opta standardises pitch dimensions to 68m x 120m through its coding software so these dimensions were used for this study.

The two fixed zone size models used zone sizes of 5m x 5m and 10m x 10m [3]. When creating the zones for these models, the opposition try areas was removed from the 68m x 120m pitch to leave a 68m x 110m area. This was necessary as the opposition try area is an area of implicitly high value, where a team is likely to attempt to ground the ball for a try before the next play begins. The 68 x 110m area was split into fourteen ~5m columns and twenty two 5m rows, resulting in 308 ~5m x 5m zones (EPV-308), and seven ~10m columns and eleven 10m rows, resulting in 77 ~10m x 10m zones (EPV-77). In both models, the columns closest to the touchlines were 1m narrower than all other columns. This transformation was necessary as Opta only provides location to the nearest metre, so splitting the 68m pitch width equally into 4.85m or 9.7m columns would provide no additional detail and would be more difficult to understand in practice.

In preparation for use within the EPV models, match event data were split into attacking sets. An attacking set was coded as a sequence of plays, which began when a team obtained possession of the ball and ended when the team lost possession of the ball (i.e. due to an error, handover, field kick, penalty, drop goal or try). The 59,233 plays used in this study were therefore grouped into 10,156 attacking sets (median length 4 plays per attacking set, range 1–26 plays). Table 1 provides a list of the events, which could end an attacking set and defines them.

**Table 1. List of events, which could end an attacking set.**

| Event | Description |
| --- | --- |
| Handover | A completed sixth tackle by the opposition team |
| Kick at goal | Conversion, Penalty Goal or Drop Goal Attempt |
| Foul | Any foul resulting in the opposition team receiving the ball (e.g. conceding a penalty) |
| Misplaced pass | A pass or tap down, which is intercepted by the opposition. |
| Misplaced kick | Any kick not caught by the team in possession, including bombs/grubber kicks and positional kicks |
| Handling error | Any situation where the ball is lost from the player's possession or dropped (e.g. lost in contact, dropped catch) |

**NB:** Any situation where a pass/kick missed its target player, but was not successfully collected by the opposition resulted in the continuation of the attacking set for the attacking team.

For every attacking set, the location at the beginning of each play (as a zone) was used. Therefore, each attacking set consisted of a sequence of zones equal to the number of plays in the possession. To enable the EPV models to calculate zone values, each zone visited was assigned a reward based on the outcome of the play: converted try scored (+6); unconverted try scored (+4); penalty goal scored (+2); drop goal scored (+1); loss of possession or missed goal attempt (0). In plays where none of these events occurred, a reward of 0 was assigned. Each time step within the sequence was assigned a reward, so it was possible for a zone to receive multiple rewards if more than one play began in a given location within the same attacking set.

## Calculation of EPV-308 and EPV-77 fixed zone size values

To evaluate the EPV for each zone on the pitch within the fixed zone size models, attacking sets were considered as Markov Chains whereby the location of the ball on the pitch (i.e. its zone) at a given time (i.e. play) within the possession was represented as an event. The value for each zone $s$, at a given time within the possession $t$ was defined as:

$$V(s, t) = E[\sum_{k=0}^{\infty} \gamma^k R_{t+k+1} | S_t = s] \tag{1}$$

where $V(s, t)$ is the estimated reward obtained from the zone $s$, at the time $t$, $R_u = R_{t+k+1}$ is the reward obtained at the time $u$, which is determined by the end of play $u\text{-}1$ (e.g. if a converted try is scored at play u-1, $R_u = 6$), $\gamma$ is the discount factor and $k$ is a play within the attacking set.

Subsequently, the overall return of any zone $s$ after play $t$ across the sample of attacking sets was calculated as:

$$G(s, t) = \sum_{j \in A_{s,t}} \gamma^{\tau_j - 1 - t} R^j \tag{2}$$

where $G(s, t)$ refers to the overall return for zone $s$ after play $t$ across the sample of attacking sets; $A_{s,t}$ is a set of attacking sets where the ball is at location $s$, in play $t$; $\tau_j$ is the play number within an attacking set $j$, $R^j$ is the reward for the attacking set $j$.

Finally, the expected possession value (EPV) of each zone $s$ after time $t$ was simulated using the Monte Carlo every visit algorithm. The Monte Carlo every visit algorithm calculates the empirical mean of each zone by summing the discounted rewards accumulated by the zone and dividing them by the total number of visits. The algorithm allows every visit to a zone to be valued, which is important within rugby league as there is no guarantee that a play will be able to move between states as the opposition defence aims to stop progression up the pitch. It is calculated as follows:

$$EPV(s, t) \approx \frac{1}{|A_{s,t}|} G(s, t) \tag{3}$$

## Calculation of aggregated zone values (EPV-37, EPV-19, EPV-13, EPV-9)

To calculate the aggregated set of zones for EPV-37, EPV-19, EPV-13 and EPV-9, the zones from EPV-308 were grouped together or split based upon differences in their match EPV. The match EPV ($G_m(s,t)$) for zone $s$ in match $m$ was calculated using Eq 2. However, rather than considering the zones' match EPV individually, they were summed at a column level to provide the column match EPV, and at a row level to provide the row match EPV. This allowed the influence of, for example, starting a play in a wide location to be evaluated globally across the pitch.

Visual inspection of the initial column and row values showed that they could be smoothed based on their spatial similarity. The fourteen columns were averaged at the 5m level symmetrically to form seven ~10m columns, whereas the twenty two 5m rows were aggregated to

eleven 10m rows similar to Kempton et al. (2016). Following this initial aggregation, linear mixed models were used to evaluate whether the columns or rows could be further combined. In separate models, the column match EPV and row match EPV were added as dependent variables, with team and fixture ID added as random effects. To evaluate for differences in their match EPVs, column and row indexes were added to their respective models as categorical fixed effects. Minimal effects testing [12] was used in four separate models to determine whether two columns or rows could be combined against a smallest effect size of interest (SESOI) of 0.5, 1.0, 1.5 and 2.0 units of match EPV respectively. If the difference between two columns or rows was statistically significant (i.e. $P < 0.05$), they remained separate; otherwise, their column/row match EPVs were averaged and compared to the next column or row's match EPV. This iterative process was conducted independently for the columns and rows. The columns and rows were then combined to form a grid. All zones were aggregated at a row level between -10m and 10m, rather than splitting them into columns. This decision was made because the zones within the row were visited infrequently relative to the other zones and so had highly variable zone values. All statistical values obtained within this process are provided in S1 File.

At SESOI 0.5, 7 rows and columns were present, resulting in 37 zones for the EPV-37. At SESOI 1.0, 4 rows and 6 columns were present, resulting in 19 zones for the EPV-19. At SESOI 1.5, 4 rows and 4 columns were present, resulting in 13 zones for the EPV-13. At SESOI 2.0, 3 rows and 4 columns were present, resulting in 9 zones for the EPV-9. The aggregated zone values were calculated as a weighted average of the values of the EPV-308 zones they were composed of. Fig 1 highlights this process by depicting the similarities and differences between the EPV-308, EPV-77 and EPV-19 in the 30m closest to the opposition try line.

## Evaluating the reproducibility of match EPV between fixtures

To evaluate the reproducibility of match EPV between fixtures, the individual zones' match EPV was compared between fixtures at a team level. The previous 1–10 fixtures were compared against the subsequent fixture. Every possible fixture within the 2019 Super League season was evaluated, resulting in 28 comparisons per team when one previous fixture was considered through to 19 comparisons per team when ten previous fixtures were considered.

The Kullback-Leibler (KL) Divergence [13] was used to calculate the similarity between the reward distribution in the subsequent match $i$ and the reward distribution in the $k$ previous match(es). The reward distribution for zone $s$ in match $mi$ ($PG_{mi}(s)$) was calculated via the equation:

$$PG_{mi}(s) = \frac{G_{mi}(s)}{\sum_{s=1}^{S} G_{mi}(s)} \tag{4}$$

The reward distribution for zone $s$ in $k$ matches prior to match $i$ was calculated as:

$$PG_M(s) = \frac{\sum_{k=1}^{k=i-1} G_{mk}(s)}{\sum_{k=1}^{k=i-1} \sum_{s=1}^{S} G_{mk}(s)} \tag{5}$$

where $PG_M(s)$ refers to the reward distribution obtained by zone $s$ across $M$ fixtures, $G_m(s)$ refers to the match EPV obtained by zone $s$ in match $m$ and $S$ refers to the set of all zones within the EPV model.

The KL Divergence is calculated according to the equation:

$$D_{KL}(P||Q) = \sum_{s \in S} P(s) \log\left(\frac{P(s)}{Q(s)}\right)$$

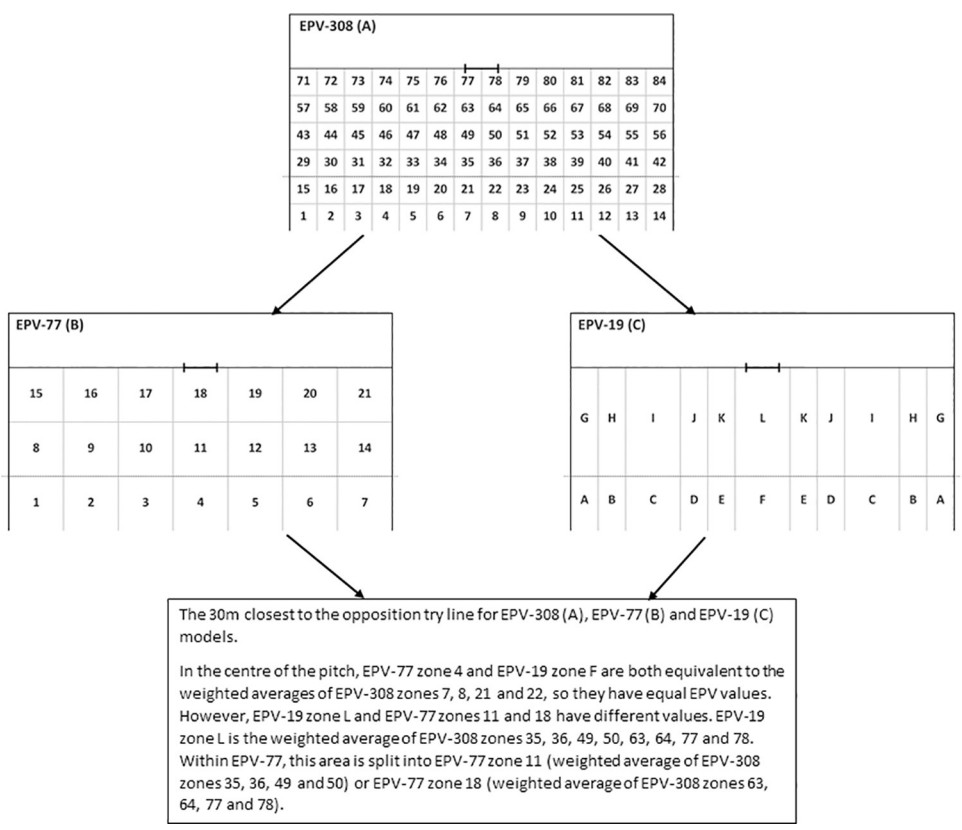

**Fig 1.** Depiction of similarities and differences between EPV-308 (A), EPV-77 (B) and EPV-19 (C) in the 30m closest to the opposition try line. Each zone from the EPV-77 and EPV-19 is a weighted average of the EPV-308 zones they are composed of. Dotted line represents the opposition 20m line.

Where P is the true reward distribution, Q is the approximating reward distribution and S refers to the set of all states within the model. The subsequent match's reward distribution ($PG_{mi}(S)$) was used as the true reward distribution with the previous matches' reward distribution ($PG_M(S)$) as the approximating distribution.

The KL Divergence is a measure used in information theory and provides an understanding of the similarity between two distributions of values. It is an unbounded measure, where a value of 0 indicates two distributions are perfectly matched, but a value of infinity indicates that there is no relationship between the two distributions. A value of infinity typically occurs when a zone within the approximating distribution ($PG_M(S)$) has no value (i.e. it has not been visited), but has been visited by the true distribution ($PG_{mi}(S)$). The percentage of non-infinity values was used to provide an understanding of how many of the subsequent match's zones were completely visited in the previous matches. The KL Divergence value was used as a measure of similarity between the two reward distributions' values. All results are provided as a mean and standard deviation values across the twelve Super League clubs.

## Quantifying teams' attacking performances using EPV-19

The attacking performances of individual teams was quantified using z-score analysis. Each team's reward distribution across the 2019 Super League season was calculated for the EPV-19 via Eq 5. Z-score analysis of the reward distributions was used to calculate a standardised value evaluating how the proportion of match EPV a team obtained from a zone compared to the

average across all teams in the Super League. Values of +1 and +2 z-scores were chosen to represent greater and much greater proportion of match EPV generated by the zones relative to the average team, values of -1 and -2 were used to represent a lower and much lower proportion of match EPV generated.

All analyses were conducted using bespoke Python scripts (Python 3.7, Python Software Foundation, Delawere, USA) or via Proc Mixed (SAS University Edition, SAS Institute, Cary, NC).

## Results

### Calculation of EPV models

Fig 2 illustrates the zone values for all six EPV models (EPV-308, EPV-77, EPV-37, EPV-19, EPV-13 and EPV-9). There is a general trend that the closer the zone is to the opposition try line, the more valuable it is. Similarly, central zones are more valuable than wider zones as

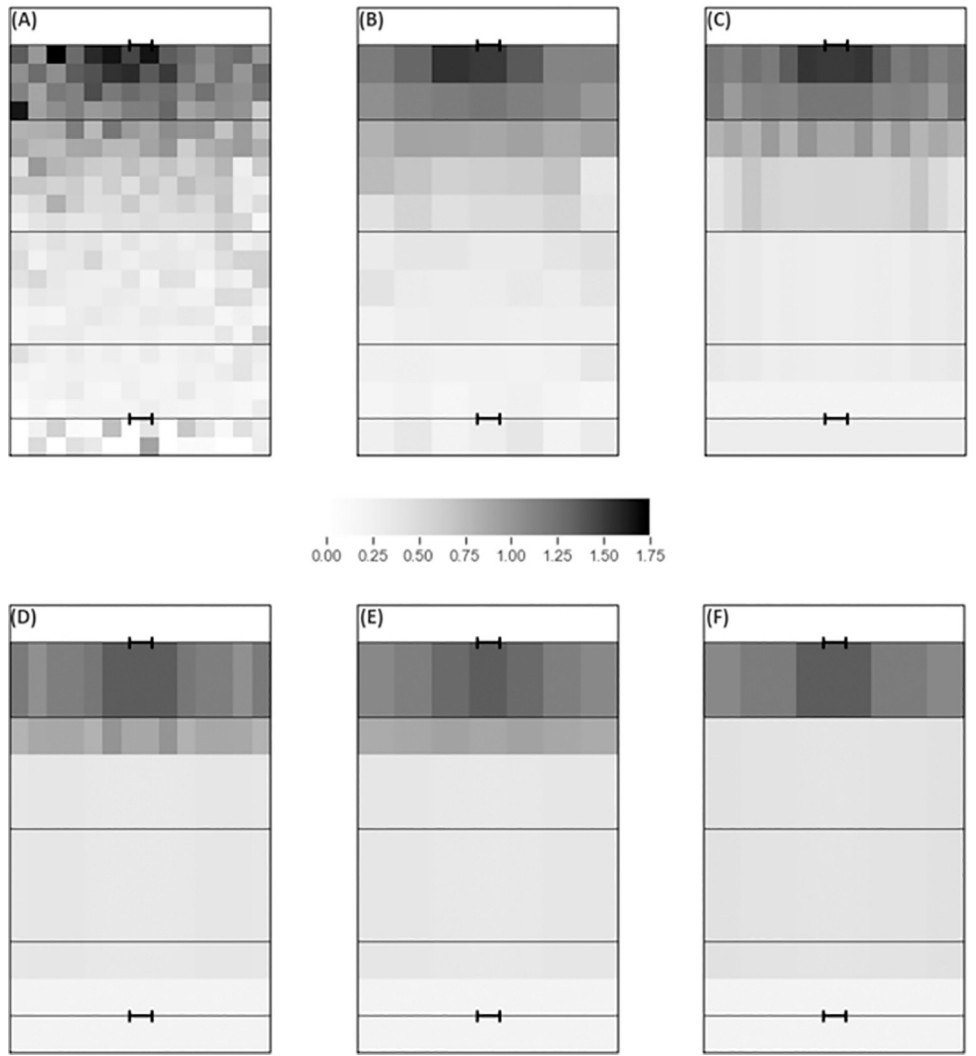

**Fig 2.** EPV-308 (A), EPV-77 (B), EPV-37 (C), EPV-19 (D), EPV-13 (E) and EPV-9 (F). Lines represent the try line, 20m line and 50m line. All state values are shaded to the same scale.

indicated by the darker colours in these areas. As the number of zones decreases, there is a smoothing effect, whereby the values of adjacent zones move closer together (indicated by the reduced light-dark contrast between them).

It is noticeable in all models that the majority of variation in values between zones is present within 30m of the opponent's try line. There are considerable differences in how each of the aggregated models handles this variation though. EPV-37 has a separate row for each 10m block, EPV-19 and EPV-13 split it into 20-30m from the try line and 0-20m from the try line, whereas EPV-9 only considers the 0-20m area to be different from the rest of the pitch.

### Reproducibility of match EPV between fixtures

Table 2 shows the percentage of non-infinity values for all six models after 1–10 previous matches (i.e. the percentage of fixtures where all subsequent match's zones had been visited in the previous matches). For EPV-308, there were only three occasions where this is not 0% (8, 9 and 10 previous matches). There was a consistent increase in the percentage of non-infinity values as the number of previous fixtures increased for EPV-77 and EPV-37, peaking at $77 \pm 8\%$ and $97 \pm 4\%$ respectively after 10 previous fixtures. For EPV-19, there was a large increase in the percentage of non-infinity values before 6 previous fixtures were considered, after which limited change was observed (95–98% from 6 to 10 fixtures). A similar trend was present for EPV-13 before 3 previous fixtures were considered (96–100% from 3 to 10 fixtures). In EPV-9, 100% of values were not infinity after only 3 previous fixtures.

Fig 3 shows the KL Divergence for EPV-77, EPV-37, EPV-19, EPV-13 and EPV-9. After 8 (KL Divergence = $1.50 \pm 0.19$), 9 (KL Divergence = $1.41 \pm 0.15$) and 10 (KL Divergence = $1.44 \pm 0.15$) previous matches, the KL Divergence for EPV-308 was greater than any other model. The KL Divergence reduced as more previous matches were considered in all EPV models in Fig 3. For EPV-37, EPV-19, EPV-13 and EPV-9, the majority of this reduction occurred between 1 and 3 previous matches before the values stabilised. For EPV-77, the values stabilised after six previous matches.

### Quantifying teams' attacking performances using EPV-19

Fig 4 provides a numbered zone breakdown for the EPV-19. Fig 5 depicts the z-score analysis of each team's attacking performances across the 2019 Super League season using the EPV-19

**Table 2. Percentage of non-infinity zones for each model, providing the percentage of matches where the complete set of the subsequent match's zones were visited in the previous *n* matches.** Values are mean (standard deviation) percentage (%) across all clubs.

| Model | Number of previous matches | | | | | | | | | |
|---|---|---|---|---|---|---|---|---|---|---|
| | 1 | 2 | 3 | 4 | 5 | 6 | 7 | 8 | 9 | 10 |
| EPV-308 | 0 | 0 | 0 | 0 | 0 | 0 | 0 | 2 | 4 | 5 |
| | (0) | (0) | (0) | (0) | (0) | (0) | (0) | (2) | (4) | (5) |
| EPV-77 | 0 | 2 | 13 | 26 | 40 | 51 | 63 | 69 | 72 | 77 |
| | (0) | (2) | (6) | (7) | (9) | (12) | (9) | (8) | (8) | (8) |
| EPV-37 | 1 | 19 | 48 | 67 | 77 | 85 | 89 | 92 | 95 | 97 |
| | (2) | (7) | (8) | (11) | (8) | (9) | (9) | (6) | (5) | (4) |
| EPV-19 | 23 | 59 | 76 | 88 | 91 | 95 | 96 | 96 | 98 | 98 |
| | (8) | (6) | (6) | (8) | (5) | (4) | (3) | (3) | (3) | (3) |
| EPV-13 | 61 | 89 | 96 | 98 | 99 | 100 | 100 | 100 | 100 | 100 |
| | (9) | (3) | (4) | (4) | (2) | (0) | (0) | (0) | (0) | (0) |
| EPV-9 | 86 | 99 | 100 | 100 | 100 | 100 | 100 | 100 | 100 | 100 |
| | (5) | (2) | (0) | (0) | (0) | (0) | (0) | (0) | (0) | (0) |

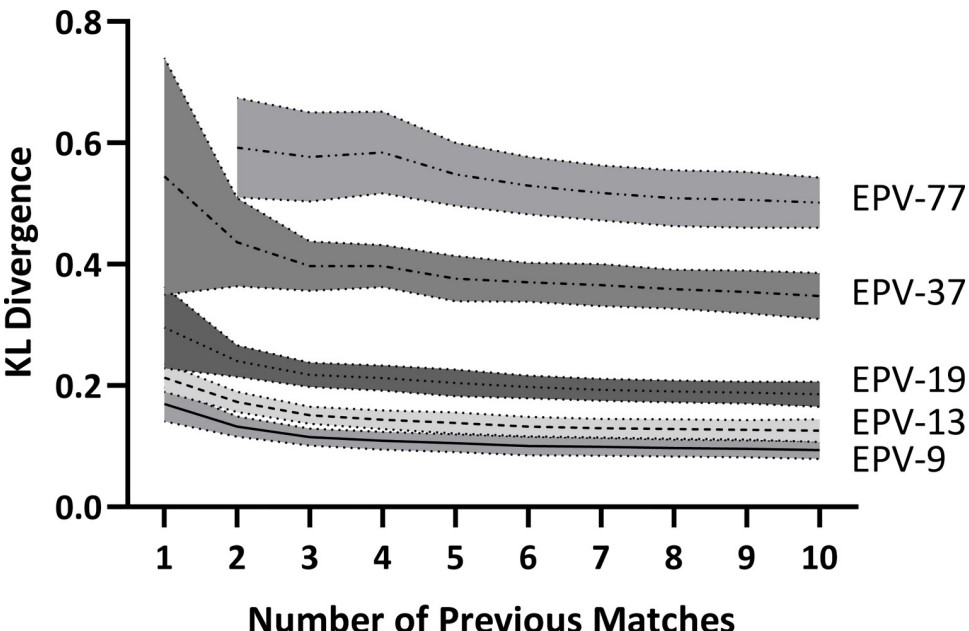

**Fig 3. KL Divergence values for EPV-77, EPV-37, EPV-19, EPV-13 and EPV-9.** A lower value indicates greater similarity in reward distributions between previous matches and the subsequent match. EPV-308 not included as values could not be calculated for the first seven matches due to no non-infinity values being present. Line provides mean value, shaded area indicates standard deviation.

model. Relative to the average Super League team, team 4 gained greater proportion of match EPV from wider areas 10-70m from their try line. Conversely, teams 6 and 8 gained a greater proportion of match EPV attacking centrally (zones 5–7 for team 6, zones 5–6 for team 8). Within 20m of the opposition try line, team 9 gained a lower proportion of match EPV from the widest zone (zone 14), but the spread of their match EPV over the more central areas was much more even than other teams.

## Discussion

The aims of this study were to i) produce six EPV models (two with fixed zone sizes of ~5m x 5m and ~10m x 10m [3], and four with aggregated zones based on differences in the zones' match EPV of 0.5, 1.0, 1.5 and 2.0 points per match) analysing attacking performance in rugby league, ii) compare the reproducibility of match EPV between fixtures for the EPV models, and iii) quantify individual teams' attacking performances across a season using an EPV model. Six EPV models were produced: EPV-308, EPV-77, EPV-37, EPV-19, EPV-13 and EPV-9. The results show that the attacking performances of previous matches, as assessed by the match EPV, were more reproducible in subsequent matches as the number of zones in the model decreased and the number of previous matches used increased. However, as reproducibility increased, the homogeneity of the zone values also increased. The results also showed that z-scores of the reward distribution could be used to identify zones through which teams obtain a greater or lower proportion of their match EPV relative to the average Super League team.

### Generation of EPV models

By generating six EPV models, it is possible to compare the value that each model estimates to be generated by possessing the ball in any location on a rugby league pitch. In all six EPV

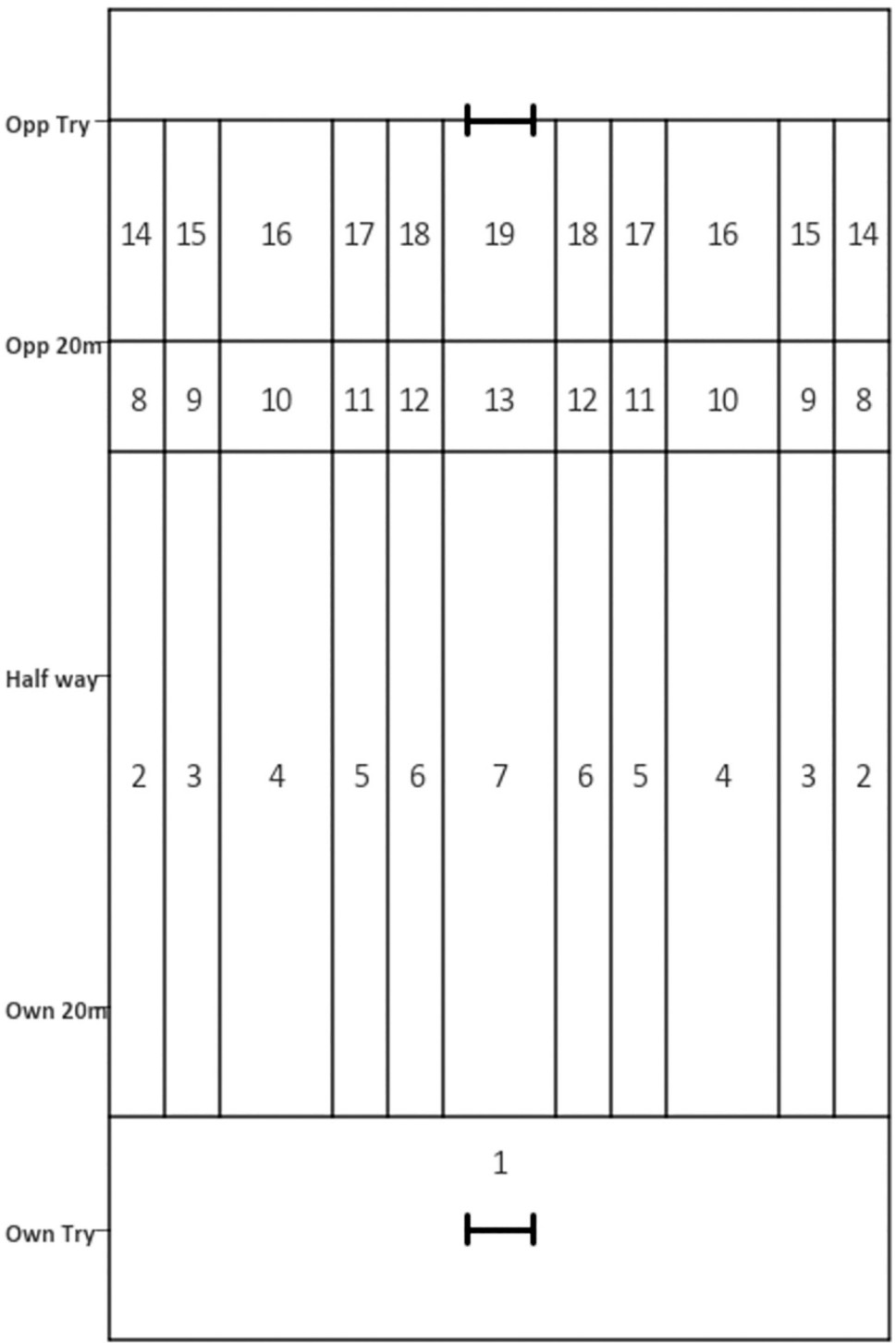

**Fig 4. EPV-19 zones numbered so they can be distinguished from each other.** Where numbers are repeated, both sides of the pitch make up the same zone (e.g. zone 2 is comprised of the widest ~5m on both sides of pitch, between 10m and 70m from the team's own try line).

models, zones were more valuable the closer they were to the opposition try line and the more central they were, which aligns with the findings of Kempton et al. [3]. Additionally, in all six models much greater value is generated within 20-30m of the opposition try line, compared

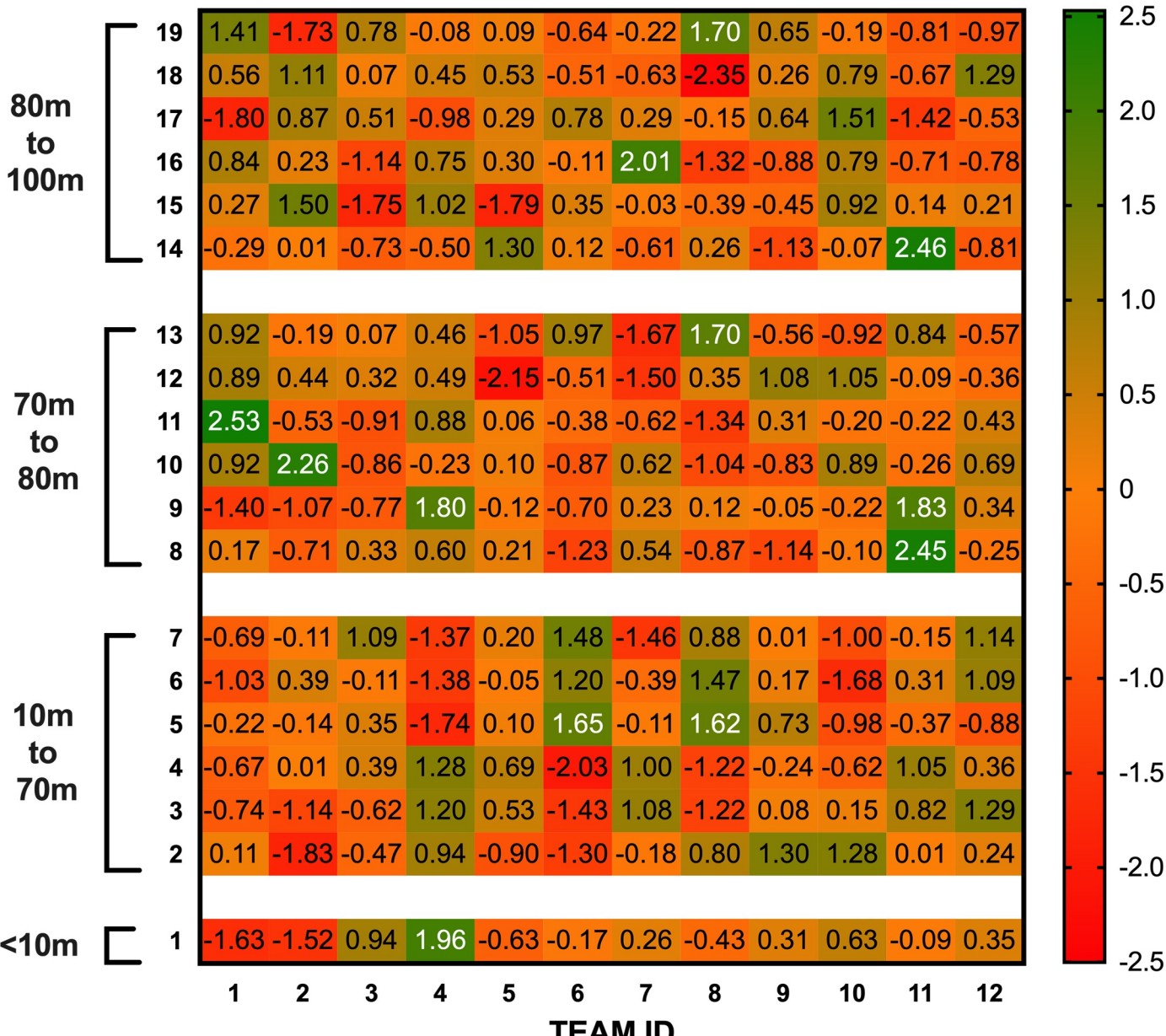

**Fig 5. Z-score analysis of teams' attacking performances in 2019 Super League season.** Numbers 1–19 reflect the zone numbers in Fig 4. A greater value indicates a greater proportion of EPV was obtained from the zone than the average Super League 2019 team. Distances are measured from the team's own try line.

with >30m. This finding is similar to previous research within football, which shows that the chance of scoring is significantly reduced to below 7% when shots are taken from outside the 18 yard box [14]. The identification of these zones of value in all six models provides a new method through which individual possessions can be valued. Furthermore, they provide a valuable methodology through which the zones visited in tactical set plays could be measured to establish which play may be most advantageous against a given team. However, it should be noted that within Kempton and colleagues' [3] model, the increase in value was much more gradual from the team in possession's 20m line through to the opposition try line than in the models produced here. Whether this is due to the methodological differences (e.g. the different

definition of possessions) or a difference in playing style between National Rugby League and Super League teams is unclear.

## Reproducibility of attacking performances

A key element of any model evaluating attacking performances is to identify how well they relate to performance in future fixtures. Despite this, few studies have attempted to evaluate this component of their models [15]. This study was the first to evaluate the reproducibility of the EPV models between fixtures via the KL Divergence. The results showed that although the EPV-308, EPV-77 and EPV-37 provide significantly more variability than either the EPV-19, EPV-13 and EPV-9 with regards to the values of different zones, they had poor reproducibility between fixtures. This was noticeable in both the percentage of subsequent match zones visited and in the similarity in reward distributions between the previous and subsequent matches. The EPV-308, EPV-77 and EPV-37 therefore have limited application in practice. By contrast, the EPV-19, EPV-13 and EPV-9 all showed excellent reproducibility between fixtures. When six previous matches were considered, these three models were able to visit all zones in the subsequent match on 95–100% of occasions. Furthermore, the reward distributions had low KL Divergence values indicating that proportion of points obtained from each zone was also very similar to the subsequent match.

Six matches is a relatively small number of matches to consider given the excellent reproducibility shown, suggesting that any of the EPV-19, EPV-13 or EPV-9 models could be used to evaluate team attacking performance in rugby league. However, it is the usefulness of the zones generated that should define which model is used in practice. The EPV-19 and EPV-13 both contain four rows (-10 to 10m, 10-70m, 70-80m, 80-100m), whereas the EPV-9 only contains three rows (-10m to 10m, 10-80m, 80-100m). As five of the six models produced suggest that the value of zones in the 70-80m row can be differentiated from those around it, it is possible that the EPV-9 has oversmoothed the data, reducing its usefulness in practice. The EPV-19 and EPV-13 models only differ in the manner through which they split group the columns along the x-axis. The EPV-19 has more columns (6), separating out the widest and second most central areas of EPV-13. This results in the EPV-13 having a smoother progression of zone values from wide to central. However, it also results in the value of the zones just outside the posts being much smaller relative to EPV-19 and EPV-37. Given the value of the central zones being of upmost importance for conversions of tries, the EPV-19 may be considered the more useful set of zones, but either model could be used in practice.

## Quantifying teams' attacking performances using EPV-19

This study used the EPV-19 model to quantify the attacking performances of individual teams within the 2019 Super League season. Using z-score analysis of the match EPV, it is clear that team 4 generates a greater proportion of match EPV than the average team from different zones to team 6 when 10m-70m from its own try line (Fig 5). Team 4 gains greater match EPV from wide areas (zones 3 and 4), whereas Team 6 gains more match EPV centrally (zones 5–7) relative to other zones. The identification of these zones pre-match could assist teams in their tactical preparations. Furthermore, the figure shows those teams who spread their attack more evenly. For example, from 80-100m, team 9 obtained a small proportion of its match EPV from the widest zone (14) compared to other Super League teams, but they generated similar proportions of match EPV across the rest of the zones. It is possible that this ability to generate value close to the average team across the majority of the pitch made the team difficult to defend against and could explain why they were one of the top points scorers across the season. The use of this z-score analysis has strong potential as a method through which the areas on a

pitch where an opposition team may attack can be highlighted quickly and efficiently, regardless of the EPV model used, enabling tactical preparations for future matches to be tailored to the opposition.

## Limitations

The EPV-19 provides an excellent starting point through which the tactical analysis of opposition teams can be conducted in a time efficient and easily interpretable manner in rugby league. However, it is subject to several limitations. The first of these is the use of only the start location of each play. Although it is not currently possible to provide any further information due to the limitations of the data used, it is important to note that the model could be improved if specific actions (e.g. passes, kicks or tackles) and their locations were included, alongside the locations of all players on the pitch. In soccer for example, every single action is location-coded by several providers, so a more complete model could be completed in that domain using a similar event level data only process. A second limitation of the model is that it does not analyse whether being aware of or attempting to stop an opposition team from visiting their highest valued zones during their attacking sets is detrimental to their ability to win rugby league matches. Future studies could resolve this by building on our framework and evaluating whether there is a difference in the zones visited when a team wins or loses matches. Similarly, future studies may wish to consider identifying specific sequences of play, as has recently been published in rugby union [16], as these sequences may also help with tactical preparations for future matches. Finally, our study also does not attempt to directly predict future attacking trends. The authors do not consider this to be a limitation due to the variability inherent within predicting single matches, but it should be highlighted. By using the KL Divergence with the subsequent match as the true distribution, any areas with 0 values in the subsequent match are automatically believed to have 0 values in the approximating distribution. Consequently, teams could spend time preparing for the opposition to use a zone they have used in previous matches, which they don't end up visiting in this specific match. However, using the EPV-19 model, the team would be prepared for the vast majority of other zones that the opponent visits during attacking sets based on the previous six matches' performances.

## Conclusions

In conclusion, this study produced six EPV models, which could be used to analyse a team's attacking performances in rugby league. The EPV-19 and EPV-13 both provide a useful understanding of attacking performances, which are reproducible in subsequent fixtures, when six previous matches are evaluated. Furthermore, z-score analysis comparing the proportion of match EPV generated by each zone relative to other teams within the league highlights the zones a team gains more value from and may provide a method through which the tactical preparation of rugby league teams could be enhanced.

## Supporting information

**S1 File.**
(DOCX)

## Author Contributions

**Conceptualization:** Thomas Sawczuk.

**Formal analysis:** Thomas Sawczuk.

**Methodology:** Thomas Sawczuk, Anna Palczewska.

**Supervision:** Anna Palczewska, Ben Jones.

**Visualization:** Ben Jones.

**Writing – original draft:** Thomas Sawczuk.

**Writing – review & editing:** Thomas Sawczuk, Anna Palczewska, Ben Jones.

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
