## [Decision Letter · Decision Letter 0]

10 Aug 2021

PONE-D-21-20054

Development of an expected possession value model to analyse team attacking performances in rugby league

PLOS ONE

Dear Dr. Sawczuk,

Thank you for submitting your manuscript to PLOS ONE. After careful consideration, we feel that it has merit but does not fully meet PLOS ONE’s publication criteria as it currently stands. Therefore, we invite you to submit a revised version of the manuscript that addresses the points raised during the review process.

The reviewers have some concerns about the rigour of the research which will require particular attention.

We look forward to receiving your revised manuscript.

Kind regards,

Caroline Sunderland

Academic Editor

PLOS ONE

2. Please provide additional details regarding participant consent. In the Methods section, please ensure that you have specified (1) whether consent was informed and (2) what type you obtained (for instance, written or verbal). If your study included minors, state whether you obtained consent from parents or guardians. If the need for consent was waived by the ethics committee, please include this information

Reviewers' comments:

Reviewer's Responses to Questions

**Comments to the Author**

1. Is the manuscript technically sound, and do the data support the conclusions?

Reviewer #1: No

Reviewer #2: Yes

2. Has the statistical analysis been performed appropriately and rigorously? 

Reviewer #1: No

Reviewer #2: Yes

3. Have the authors made all data underlying the findings in their manuscript fully available?

Reviewer #1: No

Reviewer #2: No

4. Is the manuscript presented in an intelligible fashion and written in standard English?

Reviewer #1: Yes

Reviewer #2: Yes

5. Review Comments to the Author

Reviewer #1: This study investigated the validity of the evaluation method of team attacking performances in rugby league via expected possession value (EPV) models. The authors examined three EPV models (two with fixed zone sizes of ~5m x 5m and ~10m x 10m [Kempton et al. 2016, JSS], and one with aggregated zones based on the zones’ match EPV) analyzing attacking performance in rugby league. They identified the EPV model which provides the greatest reproducibility of match EPV between fixtures, and quantified individual teams’ attacking performances across a season using an EPV model.

The motivations in this paper were clear and there were some contributions (to investigate the validity of EPV with 59,233 plays in 180 Super League matches), whereas the novelty was not so high (but this seems to be no problem in this journal). My major concern is that only one spatiotemporal resolution (EPV-19) is proposed and the results cannot conclude EPV-19 is the best. At least resolutions less than 77 and less than 19 may be required to conclude it. Furthermore, the interpretation of KL divergence may be wrong and the related part should be revised (for detail, see comment #5). For these reasons (including the comments below), the manuscript may be acceptable for me if the authors can solve these problems.

Specific comments

1. In the introduction, other EPV studies (e.g., soccer and deep learning approach) should be referred to. For example:

[1] J Fernández, L Bornn, D Cervone, Decomposing the Immeasurable Sport: A deep learning expected possession value framework for soccer, MIT SSAC, 2019

[2] J Fernández, L Bornn, D Cervone, A framework for the fine-grained evaluation of the instantaneous expected value of soccer possessions, Machine Learning, 2021

2. In the introduction, other rugby studies or methods related to movements and scores should be referred to. For example, see Related work in:

Rory Bunker, Keisuke Fujii, Hiroyuki Hanada, Ichiro Takeuchi, Supervised sequential pattern mining of event sequences in sport to identify important patterns of play: an application to rugby union, 2020.

3. The above of Eq.(3): I am not familiar with Monte Carlo every visit algorithm. Please explain the summary of the algorithm and the reason why the authors selected it.

4. P7: the authors used linear mixed models to evaluate whether the columns or rows could be combined. However, there were no related statistical values. I wonder how the areas of EPV-19 were determined based on such an approach. A detailed and clear flow of the determination may be required for reproducibility.

5. P9: KL divergence is an asymmetric measure between probability distributions P and Q. In this case, what are P and Q? These should be specified with the definition of KL divergence (i.e., equation). By the usual definition, when Q(x)=0, the KL divergence will be an infinite value. This does not always mean P (or Q) is reproducible by Q (or P). In summary, I consider that the authors may misunderstand the interpretation of KL divergence about reproducibility (this is to measure the dissimilarity between two probability distributions). This point should be revised throughout the manuscript.

6. Results and Discussion: although the analysis using KL divergence may be useful in this study, the property in the similar reward distribution between previous and current matches may not be useful to extract meaningful information. For example, coarser resolution (e.g., EPV-5) may obtain more similar distributions to the previous matches, but more general and unmeaningful insight will be obtained. EPV-19 may be an appropriate resolution to obtain meaningful insight (e.g., Fig 4), but there were no related results and discussion. The analysis will enhance the validity of the authors' claims.

Reviewer #2: The authors aim to use expected possession value (EPV) models to assess attacking performance in Rugby League. I agree that the event-level and spatial data used in this paper has good value and presents interesting opportunities for AI/statistical methods to improve match analysis in Rugby. Similar datasets have led to some very successful and interest papers at top AI venues when applied for football/soccer.

It would be nice to see a more in-depth introduction and literature review about work in this space and valuing actions in sports data. This would help make the work more accessible to a wider audience who may not be as familiar with EPV models or Rugby League. A comparison in the differences between this work and that in [5] for football would also be of interest.

I think the zone selection criteria the authors have used makes sense. However, it would have been nice to have seen a visual representation of these zones in the paper and potentially an experiment comparing the different approaches for zone selection, events that occur in those zones and how this would affect their model performances.

In the data-processing it would be good to see a full-list of events that can end a sequence and then it would be interesting to explore the impact of negative reward-weights based on loss of possession at the end of a sequence vs other 0 events such as (penalties, scrums, kicks etc).

Looking at the results to compare the 3 EPV models in Figure 2 - it seems like EPV-208 (A) may be using zones that are too granular and therefore does not have enough data to fully learnt the value of some zones (e.g., a darker zone behind the teams posts). EPV-77 (B) shows some better findings with large zones and EPV-19 (C) shows the most useful of results however I would expect most coaches were already aware of the findings. However, I think the charts shown in Figure 2 would be extremely valuable for a defensive coach when planning against an opposition as he/she would be able to understand where attacks are most dangerous.

There could be more details around each of the models. The work in [5] showing an EPV model for football goes into a lot more detail around a single EPV model and I think this paper could benefit from a more detail throughout. I think it is important for the authors and reader to recognise the limitations of their work due to data etc. Finally, I think greater discussion into future work and real-world impact of the study could be beneficial.

It would be good if eventually there is a free online source of the data used for a sample number of games in Rugby League as this would help with reproducibility and improvements of the models.

Overall, I think the flow of the paper could be improved to help with readability but the work describes an interesting and sound evaluation of EPV models in Rugby League. The paper does report original research and with a few tweaks it will make a valid contribution to the base of academic knowledge in sports analytics.

6. PLOS authors have the option to publish the peer review history of their article (what does this mean?). If published, this will include your full peer review and any attached files.

Reviewer #1: No

Reviewer #2: No

---

## [Author Response · Author response to Decision Letter 0]

24 Sep 2021

** THIS IS THE SAME AS THE WORD DOCUMENT ATTACHED TO THE SUBMISSION **

Comments to the Author

1. Is the manuscript technically sound, and do the data support the conclusions?

Reviewer #1: No

Reviewer #2: Yes

2. Has the statistical analysis been performed appropriately and rigorously?

Reviewer #1: No

Reviewer #2: Yes

3. Have the authors made all data underlying the findings in their manuscript fully available?

Reviewer #1: No

Reviewer #2: No

The data that was used for this study was acquired from a third-party, formerly Opta Sports, now Stats Perform. It is available from www.optaprorugby.com. The data was provided under a license agreement with Opta Sports/Stats Perform, and the data is subject to an approved research ethics application from our University. The terms of our license agreement prevent us from sharing the raw data we used for this analysis. Our ethical approval also prevents us from sharing any data in any way that could be re-identified. The metadata and (fixture/location/action) data itself would allow someone else to re-identify fixtures, teams and/or players, breaching the ethics approval given. However, it should be possible to obtain access to the data by contacting Stats Perform (www.statsperform.com/contact/). The authors had no special access privileges and all data is taken from the 2019 season of the Super League. The corresponding author is happy to liaise with any researchers who have queries about how to obtain the data.

4. Is the manuscript presented in an intelligible fashion and written in standard English?

Reviewer #1: Yes

Reviewer #2: Yes

5. Review Comments to the Author

Reviewer #1: This study investigated the validity of the evaluation method of team attacking performances in rugby league via expected possession value (EPV) models. The authors examined three EPV models (two with fixed zone sizes of ~5m x 5m and ~10m x 10m [Kempton et al. 2016, JSS], and one with aggregated zones based on the zones’ match EPV) analyzing attacking performance in rugby league. They identified the EPV model which provides the greatest reproducibility of match EPV between fixtures, and quantified individual teams’ attacking performances across a season using an EPV model.

The motivations in this paper were clear and there were some contributions (to investigate the validity of EPV with 59,233 plays in 180 Super League matches), whereas the novelty was not so high (but this seems to be no problem in this journal). My major concern is that only one spatiotemporal resolution (EPV-19) is proposed and the results cannot conclude EPV-19 is the best. At least resolutions less than 77 and less than 19 may be required to conclude it. Furthermore, the interpretation of KL divergence may be wrong and the related part should be revised (for detail, see comment #5). For these reasons (including the comments below), the manuscript may be acceptable for me if the authors can solve these problems.

Specific comments

1. In the introduction, other EPV studies (e.g., soccer and deep learning approach) should be referred to. For example:

[1] J Fernández, L Bornn, D Cervone, Decomposing the Immeasurable Sport: A deep learning expected possession value framework for soccer, MIT SSAC, 2019

[2] J Fernández, L Bornn, D Cervone, A framework for the fine-grained evaluation of the instantaneous expected value of soccer possessions, Machine Learning, 2021

Thank you for this comment, we have now added references to these methods in the introduction (INTRODUCTION: PARAGRAPH 2).

2. In the introduction, other rugby studies or methods related to movements and scores should be referred to. For example, see Related work in:

Rory Bunker, Keisuke Fujii, Hiroyuki Hanada, Ichiro Takeuchi, Supervised sequential pattern mining of event sequences in sport to identify important patterns of play: an application to rugby union, 2020.

Thank you for this comment. The paper mentioned answers a slightly different question to ours as we are looking at which zones are visited and attempting to find optimal zone sizes rather than the sequences of actions themselves. However, we do see the relevance of the paper as an alternative method and have therefore made reference to it in the discussion. Indeed one of the limitations of our method is that although we now know the value of different zones and where an opposition team is likely to attack from, we do not know the sequence of events through which this happens. This limitation has been added to the discussion, referencing the Bunker et al. study. (DISCUSSION – LIMITATIONS)

3. The above of Eq.(3): I am not familiar with Monte Carlo every visit algorithm. Please explain the summary of the algorithm and the reason why the authors selected it.

Thank you for this comment, we have added the details requested (METHODS – CALCULATION OF EPV-308 AND EPV-77 FIXED ZONE SIZE VALUES: PARAGRAPH 3).

4. P7: the authors used linear mixed models to evaluate whether the columns or rows could be combined. However, there were no related statistical values. I wonder how the areas of EPV-19 were determined based on such an approach. A detailed and clear flow of the determination may be required for reproducibility.

Thank you for this comment. We have now included every single statistical value obtained in the supplementary data for this paper. This information provides significantly more detail as to how the areas were determined (SUPPLEMENTARY DATA 1).

5. P9: KL divergence is an asymmetric measure between probability distributions P and Q. In this case, what are P and Q? These should be specified with the definition of KL divergence (i.e., equation). By the usual definition, when Q(x)=0, the KL divergence will be an infinite value. This does not always mean P (or Q) is reproducible by Q (or P). In summary, I consider that the authors may misunderstand the interpretation of KL divergence about reproducibility (this is to measure the dissimilarity between two probability distributions). This point should be revised throughout the manuscript.

Thank you for this comment. In this study, we use the subsequent match as the true distribution (P), and the previous 1-10 matches as the approximating distribution (Q). We are only interested in whether Q can approximate P as we only know Q when we are preparing for future matches. In line with your comments, we have extended our use of the KL Divergence to ensure the similarity aspect is used more. We still use the term reproducibility as a global term (this can be changed if there is preference for a different term), but now we consider both the percentage of non-infinity zones and the average value of the KL Divergence across all teams. We use the percentage of non-infinity zones is useful as a measure of how many of the subsequent match’s complete set of zones are visited in the previous matches, given a zone not visited by P (the subsequent match), automatically obtains a value of 0. The KL Divergence value then provides an understanding of how similar the reward distributions are between matches. To this end, as you note, we are looking at how similar the distributions are in terms of their values, rather than whether they have just been visited. We believe that our definition and usage now matches the more accurate definition you provided in the comment above. We introduce these elements in the methodology (METHODS – EVALUATING THE REPRODUCIBILITY OF MATCH EPV BETWEEN FIXTURES), provide the percentage of non-infinity values in TABLE 2, the KL Divergence values in FIGURE 3, and evaluate the results within the discussion (DISCUSSION – REPRODUCIBILITY OF ATTACKING PERFORMANCES).

6. Results and Discussion: although the analysis using KL divergence may be useful in this study, the property in the similar reward distribution between previous and current matches may not be useful to extract meaningful information. For example, coarser resolution (e.g., EPV-5) may obtain more similar distributions to the previous matches, but more general and unmeaningful insight will be obtained. EPV-19 may be an appropriate resolution to obtain meaningful insight (e.g., Fig 4), but there were no related results and discussion. The analysis will enhance the validity of the authors' claims.

Thank you for this comment. We agree that a coarser resolution (e.g. EPV-5) would provide more similar distributions to previous matches, but poor insight in practice. We also agree that more analyses would be beneficial to the study. As such, we have added three further zone analyses to the study. In the original manuscript, we used a smallest effect size of interest of 1 to split zones. We have now added three further state spaces (EPV-37, EPV-13, EPV-9), which use smallest effect sizes of interest of 0.5, 1.5 and 2.0 respectively to split the zones. We feel that this adds a significant amount to the article and thank you again for the suggestion.

Reviewer #2: The authors aim to use expected possession value (EPV) models to assess attacking performance in Rugby League. I agree that the event-level and spatial data used in this paper has good value and presents interesting opportunities for AI/statistical methods to improve match analysis in Rugby. Similar datasets have led to some very successful and interest papers at top AI venues when applied for football/soccer.

It would be nice to see a more in-depth introduction and literature review about work in this space and valuing actions in sports data. This would help make the work more accessible to a wider audience who may not be as familiar with EPV models or Rugby League. A comparison in the differences between this work and that in [5] for football would also be of interest.

Thank you for this comment, we have now added more detail to the introduction to make reference to the probabilistic deep learning models used in football vs the stochastic Markovian models used in ice hockey. We have also explained why rugby league is better suited to stochastic analyses. (INTRODUCTION: PARAGRAPH 2).

I think the zone selection criteria the authors have used makes sense. However, it would have been nice to have seen a visual representation of these zones in the paper and potentially an experiment comparing the different approaches for zone selection, events that occur in those zones and how this would affect their model performances.

Thank you for this comment, we are unclear what is meant by a ‘visual representation of these zones’. We believe this may be provided by Figure 2, but have also provided a non-heatmapped version of each model in the supplementary data (SUPPLEMENTARY DATA 1). 

We agree that the events that occur in any zone could have a large impact on the results. Unfortunately, we are unable to provide the events that occur in the specific zones due to limitations within the dataset (only the location of the first event of each play is provided, so we cannot guarantee the location accuracy of any event thereafter). However, we have added three further experiments regarding the mixed model approach for zone selection (using SESOI 0.5, 1.0, 1.5 and 2.0 now, vs only SESOI 1.0 before) to provide greater discussion around the models. 

In the data-processing it would be good to see a full-list of events that can end a sequence and then it would be interesting to explore the impact of negative reward-weights based on loss of possession at the end of a sequence vs other 0 events such as (penalties, scrums, kicks etc).

Thank you for this comment, we have now added a full list of events which could end an episode (TABLE 1). We considered negative reward weights for this study but were uncertain as to how we could include them given our episodes began when the team obtained possession and ended when the team lost possession. We believe negative events could only be included if the episode duration was extended (i.e. to the next points being scored), but this in turn creates problems as the model becomes more like a Markov Game. Such an approach is better suited with both larger datasets, and action data, neither of which was suitably available for this study. We agree the use of negative rewards is of great interest though and are hoping to identify an appropriate method of using them, or accounting for negative actions, in future studies.

Looking at the results to compare the 3 EPV models in Figure 2 - it seems like EPV-208 (A) may be using zones that are too granular and therefore does not have enough data to fully learnt the value of some zones (e.g., a darker zone behind the teams posts). EPV-77 (B) shows some better findings with large zones and EPV-19 (C) shows the most useful of results however I would expect most coaches were already aware of the findings. However, I think the charts shown in Figure 2 would be extremely valuable for a defensive coach when planning against an opposition as he/she would be able to understand where attacks are most dangerous.

Thank you, we agree with all these points. We believe that you are correct in the assertion that most coaches may expect to see some of the findings but visualising them simply in such a quantitative way could provide some insight in a quick way, which may help to prepare for future matches.

There could be more details around each of the models. The work in [5] showing an EPV model for football goes into a lot more detail around a single EPV model and I think this paper could benefit from a more detail throughout. I think it is important for the authors and reader to recognise the limitations of their work due to data etc. Finally, I think greater discussion into future work and real-world impact of the study could be beneficial.

Thank you for this comment. We have added significant detail to the development of the zones for each of the models in the supplementary material now, which we hope helps the reader to better understand how we have generated each model (SUPPLEMENTARY DATA 1). 

We have also added further information regarding limitations (data, lack of predictive work within this study), future work (attempts to improve data collection techniques and provide more meaningful models including action/player location data) and real-world impact (use of EPV-19 to evaluate the attacking performances of opposition teams) throughout the DISCUSSION.

It would be good if eventually there is a free online source of the data used for a sample number of games in Rugby League as this would help with reproducibility and improvements of the models.

Thank you, we agree. Hopefully, given the increasing amounts of free data available in other sports (soccer in particular), sports like rugby (league and union) will follow suit and release data so that models can be improved. Unfortunately, although the data used for this study is accessible, it is behind a paywall, as described in the data availability statement.

Overall, I think the flow of the paper could be improved to help with readability but the work describes an interesting and sound evaluation of EPV models in Rugby League. The paper does report original research and with a few tweaks it will make a valid contribution to the base of academic knowledge in sports analytics.

Thank you

---

## [Decision Letter · Decision Letter 1]

13 Oct 2021

PONE-D-21-20054R1Development of an expected possession value model to analyse team attacking performances in rugby leaguePLOS ONE

Dear Dr. Sawczuk,

Thank you for submitting your manuscript to PLOS ONE. After careful consideration, we feel that it has merit but does not fully meet PLOS ONE’s publication criteria as it currently stands. Therefore, we invite you to submit a revised version of the manuscript that addresses the points raised during the review process.

Please address the minor comments and then the paper will be ready for acceptance.

We look forward to receiving your revised manuscript.

Kind regards,

Caroline Sunderland

Academic Editor

PLOS ONE

Journal Requirements:

Additional Editor Comments (if provided):

Reviewers' comments:

Reviewer's Responses to Questions

**Comments to the Author**

1. If the authors have adequately addressed your comments raised in a previous round of review and you feel that this manuscript is now acceptable for publication, you may indicate that here to bypass the “Comments to the Author” section, enter your conflict of interest statement in the “Confidential to Editor” section, and submit your "Accept" recommendation.

Reviewer #1: All comments have been addressed

2. Is the manuscript technically sound, and do the data support the conclusions?

Reviewer #1: Yes

3. Has the statistical analysis been performed appropriately and rigorously? 

Reviewer #1: Yes

4. Have the authors made all data underlying the findings in their manuscript fully available?

Reviewer #1: No

5. Is the manuscript presented in an intelligible fashion and written in standard English?

Reviewer #1: Yes

6. Review Comments to the Author

Reviewer #1: Thanks for your response and revising the manuscript.

I consider that the manuscript is ready for publication if the authors revise the following points.

1. L167: EPV-37, EPV-19, EPV-13 and EPV-9, EPV-308 zones -> EPV-37, EPV-19, EPV-13, EPV-9, and EPV-308?

2. KL divergence formula should be clarified (P and Q are not defined in this paper).

7. PLOS authors have the option to publish the peer review history of their article (what does this mean?). If published, this will include your full peer review and any attached files.

Reviewer #1: No

---

## [Author Response · Author response to Decision Letter 1]

19 Oct 2021

Reviewer #1: Thanks for your response and revising the manuscript.

I consider that the manuscript is ready for publication if the authors revise the following points.

Thank you again for your critique during the review period. We are pleased that our changes satisfied your comments.

1. L167: EPV-37, EPV-19, EPV-13 and EPV-9, EPV-308 zones -> EPV-37, EPV-19, EPV-13, EPV-9, and EPV-308?

This sentence has now been clarified.

“To calculate the aggregated set of zones for EPV-37, EPV-19, EPV-13 and EPV-9, the zones from EPV-308 were grouped together or split based upon differences in their match EPV.”

2. KL divergence formula should be clarified (P and Q are not defined in this paper).

We have now added the KL Divergence formula to the text, with more description to ensure there is no ambiguity between P and Q, our true and approximating distributions.

“The KL Divergence is calculated according to the equation:

D_KL (P||Q)= ∑_(s∈S)▒〖P(s) log⁡((P(s))/(Q(s))) 〗

Where P is the true reward distribution, Q is the approximating reward distribution and S refers to the set of all states within the model. The subsequent match’s reward distribution (PGmi(S)) was used as the true reward distribution with the previous matches’ reward distribution (PGM(S)) as the approximating distribution. “

---

## [Editor Report · Decision Letter 2]

21 Oct 2021

Development of an expected possession value model to analyse team attacking performances in rugby league

PONE-D-21-20054R2

Dear Dr. Sawczuk,

We’re pleased to inform you that your manuscript has been judged scientifically suitable for publication and will be formally accepted for publication once it meets all outstanding technical requirements.

Kind regards,

Caroline Sunderland

Academic Editor

PLOS ONE
---

## [Editor Report · Acceptance letter]

3 Nov 2021

PONE-D-21-20054R2 

Development of an expected possession value model to analyse team attacking performances in rugby league 

Dear Dr. Sawczuk:

I'm pleased to inform you that your manuscript has been deemed suitable for publication in PLOS ONE. Congratulations! Your manuscript is now with our production department. 

Kind regards, 

on behalf of

Dr. Caroline Sunderland 

Academic Editor

PLOS ONE